# Classical and Relativistic Evolution of an Extra-Galactic Jet with Back-Reaction

**Lorenzo Zaninetti** 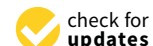

Department of Physics, University of Turin, via P.Giuria 1, I-10125 Turin, Italy; zaninetti@ph.unito.it

**Abstract:** We consider a turbulent jet that is moving in a Lane–Emden ($n = 5$) medium. The conserved quantity is the energy flux, which allows finding, to first order, an analytical expression for the velocity and an approximate trajectory. The conservation of the relativistic flux for the energy allows deriving, to first order, an analytical expression for the velocity, and numerically determining the trajectory. The back-reaction due to the radiative losses for the trajectory is evaluated both in the classical and the relativistic case.

**Keywords:** radio galaxies; jets and bursts; galactic winds and fountains; radio sources

## 1. Introduction

The study of extra-galactic jets started with the observations of NGC 4486 (M87), where "a curious straight ray lies in a sharp gap in the nebulosity ..."; see [1] and Figure 1.

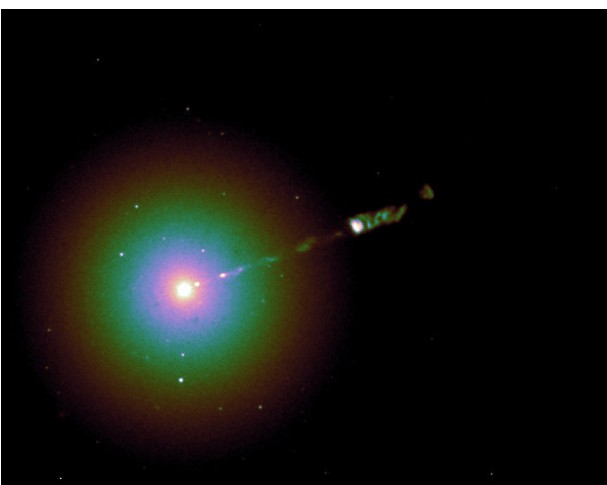

**Figure 1.** The super-giant elliptical galaxy M87 and the optical jet; the credit is due to Instituto de Astrofísica de Canarias.

At the moment of writing, the extra-galactic radio sources are classified on the basis of the position of the brightest radio-emitting regions with respect to the channel; see [2,3] for details. FR-I, after Fanaroff and Riley, have hot spots that are more distant from the nucleus (a typical example is Cygnus A) and luminosity, $L$, at 178 MHz of:

$$L \leq h_{100}\, 2 \times 10^{25} \frac{\text{W}}{\text{Hz\,str}} \quad \text{FR-I,} \tag{1}$$

where $h_{100} = H_0/100$ and $H_0$ is the Hubble constant. FR-II radio galaxies have emission uniformly distributed along the channel (a typical example is 3C449) and luminosities greater than the above value, or, in other words, the more powerful radio galaxies are classified as FR-II. A list of the properties, length in kpc, and power in Watts of extra-galactic radio jets can be found in [4,5]. In the following, we will study jets with small openings, such as that of M87.

The problem of the velocity of extra-galactic radio jets has been analyzed in two ways:

1. The velocity of the jet is constant over many kpc and takes the value $v$. Due to the fact that it is thought that this velocity is nearly relativistic, it is parametrized as $\beta = \frac{v}{c}$, where $c$ is the velocity of light. As an example, [6] analyzed some wide-angle tail radio galaxies and found a terminal velocity of $\beta = 0.3$.
2. The velocity of the jet decreases with an imposed law (see [7]) or is evaluated by a numerical code (see [8,9]). In this case, the relativistic parameter $\beta$ decreases along the trajectory.

Recently, the problem of the decrease of the velocity along a turbulent jet has been solved, imposing the conservation of the flux of momentum (see [10]) or imposing the conservation of the energy flux (see [11]). The approach using the conservation of the flux of energy is attractive because it has the same dimension of the luminosity. Further on, the jets are radiating away in the various observational bands, such as radio, optical, infrared, etc., and we briefly recall that the extra-galactic radio source covers a range in observed luminosity from $10^{19} \frac{W}{Hz}$–$10^{28} \frac{W}{Hz}$ (see [12]).

Therefore, the flux of energy available at the beginning of the jet will progressively decrease due to the radiative losses. This paper, in Section 2, introduces the Lane–Emden ($n = 5$) density profile and consequently derives an approximate trajectory to first order, as well as a numerical trajectory to second order in the presence of losses. In Section 3, we present a series solutions for the relativistic trajectory to first order and a numerical solution to second order. Section 4 models the intensity of the radio-jet in 3C31.

## 2. Conservation of the Flux of Energy

A turbulent jet is defined as a jet that has the same density as the surrounding intergalactic medium (IGM); see the next subsection for details. The conservation of the flux of energy in a turbulent jet has been explored in [11] for three types of IGM, with the following radial dependences: constant density profile, hyperbolic, and inverse power law density profiles. Here, we analyze the case of a Lane–Emden ($n = 5$) density profile, to which a subsection will be dedicated.

### 2.1. The Turbulent Jet

Turbulent jets are a subject of laboratory experiments; as an example, see Figure 2.

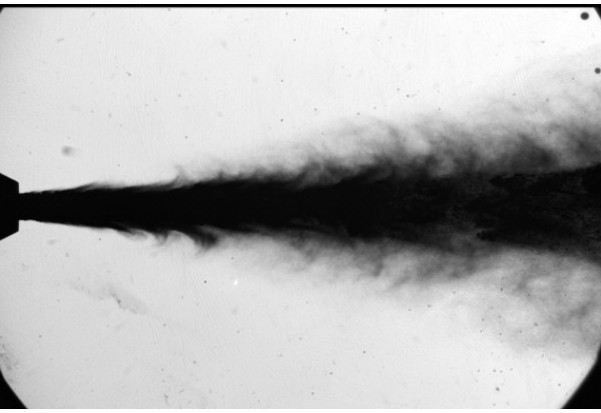

**Figure 2.** Coaxial liquid-air jet; the credit is due to Mixing Enhancement via Secondary Parallel Injection (MESPI).

The theory of turbulent jets emerging from a circular hole can be found in different books with different theories; see [13–15]. The basic assumptions common to the three already cited approaches are:

1.　The rate of momentum flow, $J$, represented by:

$$J = constant \times \rho b_j^2 \overline{v}_{x,max}^2, \tag{2}$$

is constant; here, $x$ is the distance from the initial circular hole, $b_j(x)$ is the jet's diameter at distance $x$, $\overline{v}_{x,max}$ is the maximum velocity along the the centerline, *constant* is:

$$constant = 2\pi \int_0^\infty f^2 \xi d\xi, \tag{3}$$

where:

$$f(\xi) = \frac{\overline{v}_x}{\overline{v}_{x,max}} \quad with \quad \xi = \frac{x}{b_{1/2}}, \tag{4}$$

and $\rho$ is the density of the surrounding medium; see Equation (5.6-3) in [13].

2.　The jet's density $\rho$ is constant over the expansion and equal to that of the surrounding medium. The pressure is absent in this theory.

Omitting the details of the computation, an expression can be found for the average velocity $\overline{v}_x$ (see Equation (5.6-21) in [13]),

$$\overline{v}_x = \frac{v^{(t)}}{x} \frac{2C_3^2}{\left[1 + \frac{1}{4}(C_3 \frac{r}{x})^2\right]^2}, \tag{5}$$

where $v^{(t)}$ is the kinematical eddy viscosity and $C_3$ is as follows (see Equation (5.6-23) in [13]),

$$C_3 = \sqrt{\frac{3}{16\pi}} \sqrt{\frac{J}{\rho} \frac{1}{v^{(t)}}}. \tag{6}$$

An important quantity is the radial position, $r = b_{1/2}$, corresponding to an axial velocity one-half of the centerline value (see Equation (5.6-24) in [13]),

$$\frac{\overline{v}_x(b_{1/2}, x)}{\overline{v}_{x,max}(x)} = \frac{1}{2} = \frac{1}{\left[1 + \frac{1}{4}(C_3 \frac{b_{1/2}}{x})^2\right]^2}. \tag{7}$$

The experiments in the range of the Reynolds number, $Re$, $10^4 \le Re \le 3 \times 10^6$ (see [16–18]), indicate that:

$$b_{1/2} = 0.0848x, \tag{8}$$

and as a consequence:

$$C_3 = 15.17, \tag{9}$$

and therefore:

$$\frac{\overline{v}_x(r)}{\overline{v}_{x,max}(r)} = \frac{1}{\left[1 + 0.414(\frac{r}{b_{1/2}})^2\right]^2}. \tag{10}$$

The average velocity, $\overline{v}_x$, is $\approx 1/100$ of the centerline value when $r/b_{1/2} = 4.6$, and this allows seeing that the diameter of the jet is:

$$b_j = 2 \times 4.6 b_{1/2}. \tag{11}$$

On introducing the opening angle $\alpha$, the following new relation is found:

$$\frac{\alpha}{2} = \arctan \frac{4.6 b_{1/2}}{x}. \tag{12}$$

The generally accepted relation between the opening angle and Mach number (see Equation (A33) in [19]) is:

$$\frac{\alpha}{2} = \frac{c_s}{v_j} = \frac{1}{M},$$

(13)

where $c_s$ is the velocity of sound, $v_j$ the jet's velocity, and $M$ the Mach number. The new relation (12) replaces the traditional relation (13). The parameter $b_{1/2}$ can therefore be connected with the jet's geometry:

$$b_{1/2} = \frac{1}{4.6} \tan(\frac{\alpha}{2}) x.$$

(14)

If this approximate theory is accepted, Equation (8) gives $\alpha = 42.61°$; this is the theoretical value that yields the so-called Reichardt profiles. The value of $b_{1/2}$ fixes the value of $C_3$, and therefore, the eddy viscosity is:

$$\nu^{(t)} = \sqrt{\frac{3}{16\pi}} \sqrt{\frac{J}{\rho} \frac{1}{C_3}} = \sqrt{\frac{3}{16\pi}} \sqrt{constant} \, b v_{x,max} \frac{1}{C_3}.$$

(15)

In order to continue, the integral that appears in *constant* should be evaluated; see Equation (3). Numerical integration gives:

$$\int_0^\infty f^2 \xi d\xi = 0.402,$$

(16)

and therefore:

$$constant = 2.528.$$

(17)

On introducing typical parameters of jets like $\alpha = 5°$, $v_{x,max} = v_{100} = v[\text{km/s}]/100$, $b_j = b_1$, where $b_1$ is the momentary diameter in pc, it is possible to deduce an astrophysical formula for the kinematical eddy viscosity:

$$\nu^{(t)} = 2.92 \, 10^{-9} \, b_1 v_{100} \frac{\text{pc}^2}{\text{year}} \quad when \quad C_3 = 135.61.$$

(18)

This paragraph concludes by underlining the fact that in extra-galactic sources, it is possible to observe both a small opening angle, $\approx 5°$, and great opening angles, i.e., $\approx 34°$, in the outer regions of 3C31 [20].

*2.2. The Lane–Emden Profile*

The self gravitating sphere of a polytropic gas is governed by the Lane–Emden differential equation of the second order:

$$\frac{d^2}{dx^2} Y(x) + 2 \frac{\frac{d}{dx} Y(x)}{x} + (Y(x))^n = 0,$$

where $n$ is an integer; see [21–25]. The solution $Y(x)_n$ has the density profile:

$$\rho = \rho_c Y(x)_n^n,$$

where $\rho_c$ is the density at $x = 0$. The pressure $P$ and temperature $T$ scale as:

$$P = K \rho^{1 + \frac{1}{n}},$$

(19)

$$T = K' Y(x),$$

(20)

where $K$ and $K'$ are two constants. For more details, see [26].

Analytical solutions exist for $n = 0$, 1, and 5. The analytical solution for $n = 5$ is:

$$Y(x) = \frac{1}{(1 + \frac{x^2}{3})^{1/2}},$$

and the density for $n = 5$ is:

$$\rho(x) = \rho_c \frac{1}{(1 + \frac{x^2}{3})^{5/2}}. \tag{21}$$

The variable $x$ is non-dimensional, and we now introduce the new variable $x = r/b$:

$$\rho(r; b) = \rho_c \frac{1}{(1 + \frac{r^2}{3b^2})^{5/2}}. \tag{22}$$

### 2.3. Preliminaries

The chosen physical units are pc for length and year for time; with these units, the initial velocity $v_0$ is expressed in pc year$^{-1}$. When the initial velocity is expressed in km s$^{-1}$, the multiplicative factor $1.02 \times 10^{-6}$ should be applied in order to have the velocity expressed in pc year$^{-1}$. In these units, the speed of light is $c = 0.306$ pc year$^{-1}$. The goodness of the approximation of a solution is evaluated through the percentage error, $\delta$, which is:

$$\delta = \frac{|x - x_{app}|}{x} \times 100, \tag{23}$$

where $x$ is the analytical or numerical solution and $x_{app}$ the approximate solution; see [27].

### 2.4. Classical Solution to First Order

The conservation of the energy flux in a straight turbulent jet and the concept of the perpendicular section to the motion along the Cartesian $x$-axis, $A$:

$$A(r) = \pi \, r^2 \tag{24}$$

where $r$ is the radius of the jet. The section $A$ at position $x_0$ is:

$$A(x_0) = \pi(x_0 \tan(\frac{\alpha}{2}))^2 \tag{25}$$

where $\alpha$ is the opening angle and $x_0$ is the initial position on the $x$-axis. At position $x$, we have:

$$A(x) = \pi(x \tan(\frac{\alpha}{2}))^2. \tag{26}$$

The conservation of energy flux states that:

$$\frac{1}{2}\rho(x_0)v_0^3 A(x_0) = \frac{1}{2}\rho(x)v(x)^3 A(x) \tag{27}$$

where $v(x)$ is the velocity at position $x$ and $v_0(x_0)$ is the velocity at position $x_0$; see Formula (A28) in [19]. We now assume that a Lane–Emden ($n = 5$) density profile is valid; see Equation (22). Then, the above conservation law becomes:

$$\begin{aligned}
&\frac{1}{2}\rho_0 v(x)^3 \pi \, x^2 \left(\tan\left(\frac{\alpha}{2}\right)\right)^2 \left(1 + \frac{1}{3}\frac{x^2}{b^2}\right)^{-5/2} \\
= \; &\frac{1}{2}\rho_0 v_0(x_0)^3 \pi \, x_0^2 \left(\tan\left(\frac{\alpha}{2}\right)\right)^2 \left(1 + \frac{1}{3}\frac{x_0^2}{b^2}\right)^{-5/2},
\end{aligned} \tag{28}$$

where $v(x)$ is the velocity at position $x$, $v_0(x_0)$ is the velocity at position $x_0$, and $\alpha$ is the opening angle of the jet. The above equation is a cubic equation, which has one real root plus two non-real complex conjugate roots. Here and in the following, we take into account only the real root. The real analytical solution for the velocity without losses is:

$$v(x; b, x_0, v_0) = \frac{v_0 \left(3\,b^2 + x^2\right)^{\frac{5}{6}} x_0^{\frac{2}{3}}}{\left(3\,b^2 + x_0^2\right)^{\frac{5}{6}} x^{\frac{2}{3}}}.$$

(29)

The asymptotic expansion of above velocity, $v_a$, with respect to the variable $x$, which means $x \to \infty$, is:

$$v_a(x; b, x_0, v_0) = \frac{v_0 x_0^{\frac{2}{3}} \left(5\,b^2 + 2\,x^2\right)}{2 \left(3\,b^2 + x_0^2\right)^{5/6} x}.$$

(30)

The trajectory can be found by the indefinite integral of the inverse of the velocity as given by Equation (29):

$$F(x) = \int \frac{1}{v(x; b, x_0, v_0)} dx = \frac{\sqrt[6]{3} \left(3\,b^2 + x_0^2\right)^{\frac{5}{6}} x^{\frac{5}{3}} {}_2F_1\left(\frac{5}{6}, \frac{5}{6}; \frac{11}{6}; -\frac{x^2}{3\,b^2}\right)}{5\,v_0 \left(b^2\right)^{5/6} x_0^{2/3}},$$

(31)

where ${}_2F_1(a, b; c; v)$ is a regularized hypergeometric function; see [27–30]. The trajectory expressed in terms of $t$ as a function of $x$ is:

$$F(x) - F(x_0) = t.$$

(32)

The above equation cannot be inverted in the usual form, which is $x$ as a function of $t$. The asymptotic trajectory can be found by the indefinite integral of the inverse of the asymptotic velocity as given by Equation (30):

$$F_a(x) = \int \frac{1}{v_a(x; b, x_0, v_0)} dx = \frac{\left(3\,b^2 + x_0^2\right)^{\frac{5}{6}} \ln\left(5\,b^2 + 2\,x^2\right)}{2\,v_0 x_0^{2/3}}.$$

(33)

The equation of the asymptotic trajectory is:

$$F_a(x) - F_a(x_0) = t,$$

(34)

and the solution for $x$ of the above equation, the asymptotic trajectory, is:

$$x(t; b, x_0, v_0) = \frac{1}{2}\sqrt{-10\,b^2 + 2\,e^{\frac{\left(3\,b^2 + x_0^2\right)^{5/6} \ln\left(5\,b^2 + 2\,x_0^2\right) + 2\,t v_0 x_0^{2/3}}{\left(3\,b^2 + x_0^2\right)^{5/6}}}}.$$

(35)

Figure 3 with parameters as in Table 1 shows a typical example of the above asymptotic expansion.

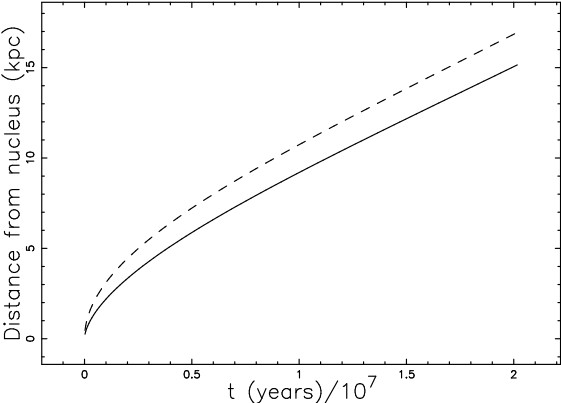

**Figure 3.** Numerical solution as given by Equation (32) (full line) and asymptotic solution as given by Equation (35) (dashed line), with parameters as in Table 1.

**Table 1.** Parameters for a classical extra-galactic jet.

| Parameter | Value |
|---|---|
| $x_0$ (pc) | 100 |
| $v_0$ ($\frac{km}{s}$) | 10,000 |
| $b$ (pc) | 10,000 |

### 2.5. Solution to Second Order

Let us suppose that the radiative losses are proportional to the flux of energy:

$$-\epsilon \frac{\rho_0 v^3 \pi x^2 \left(\tan\left(\frac{\alpha}{2}\right)\right)^2}{2\left(1 + \frac{1}{3}\frac{x^2}{b^2}\right)^{5/2}}.$$ (36)

Inserting in the above equation the velocity to first order as given by Equation (29), the radiative losses, $Q(x; x_0, v_0, b, \epsilon)$, are:

$$Q(x; x_0, v_0, b, \epsilon) = -\epsilon \frac{\rho_0 v^3 \pi x^2 \left(\tan\left(\frac{\alpha}{2}\right)\right)^2}{2\left(1 + \frac{1}{3}\frac{x^2}{b^2}\right)^{5/2}},$$ (37)

where $\epsilon$ is a constant that fixes the conversion of the flux of energy to other kinds of energies, in this case the radiative losses. The sum of the radiative losses between $x_0$ and $x$ is given by the following integral, $L$,

$$L(x; x_0, v_0, b, \epsilon) = \int_{x_0}^{x} Q(x; x_0, v_0, b, \epsilon)dx = \frac{-9\,\epsilon\,\rho_0\sqrt{3}b^5 v_0{}^3 x_0{}^2 \pi\,\left(\tan\left(\alpha/2\right)\right)^2 (x - x_0)}{2\left(3\,b^2 + x_0{}^2\right)^{5/2}}.$$ (38)

The conservation of the flux of energy in the presence of the back-reaction due to the radiative losses is:

$$\frac{9\sqrt{3}\rho_0\left(b^5 v_0{}^3 x_0{}^2 \epsilon\left(\frac{3b^2+x^2}{b^2}\right)^{5/2} x - b^5 v_0{}^3 x_0{}^3 \epsilon\left(\frac{3b^2+x^2}{b^2}\right)^{5/2} + v^3 x^2\left(3b^2 + x_0{}^2\right)^{5/2}\right)}{2\left(\frac{3b^2+x^2}{b^2}\right)^{5/2}\left(3b^2 + x_0{}^2\right)^{5/2}}$$ (39)

$$= 9\,\rho_0\sqrt{3}v_0{}^3 x_0{}^2 2\left(\frac{3b^2 + x_0{}^2}{b^2}\right)^{5/2}.$$

The analytical solution for the velocity to second order, $v_c(x; b, x_0, v_0)$, is:

$$v_c(x; b, x_0, v_0) = \frac{v_0 \sqrt[3]{1 + \epsilon \ (-x + x_0)} \left(3 \, b^2 + x^2\right)^{\frac{5}{6}} x_0^{\frac{2}{3}}}{\left(3 \, b^2 + x_0^2\right)^{\frac{5}{6}} x^{\frac{2}{3}}}, \tag{40}$$

and Figure 4 shows an example.

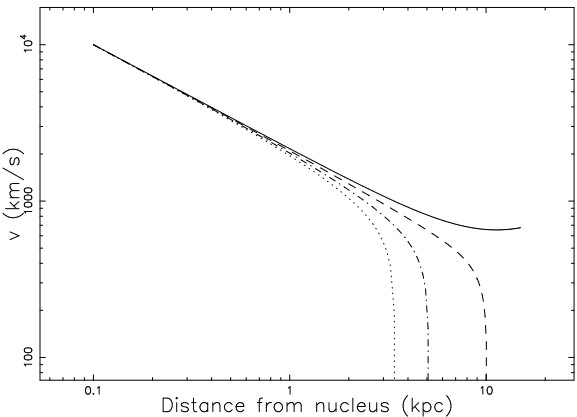

**Figure 4.** Velocity corrected for radiative losses, i.e., velocity to second order, Equation (40), as a function of the distance, with parameters as in Table 1: $\epsilon = 0$ full line, $\epsilon = 1.0 \times 10^{-4}$ dashed line, $\epsilon = 2.0 \times 10^{-4}$ dotted-dashed line, and $\epsilon = 3.0 \times 10^{-4}$ dotted line.

There are no analytical results for the trajectory corrected for radiative losses, and a numerical example is shown in Figure 5.

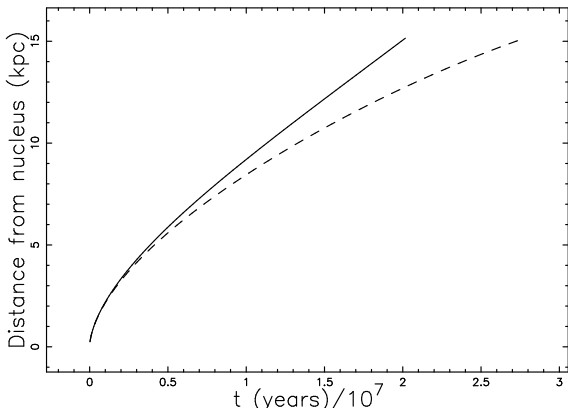

**Figure 5.** Numerical trajectory corrected for radiative losses as a function of time, with parameters as in Table 1: $\epsilon = 0$ full line and $\epsilon = 8.0 \times 10^{-5}$ dashed line.

The inclusion of back-reaction allows the evaluation of the jet's length, which can be derived from the minimum in the corrected velocity to second order as a function of $x$,

$$\frac{\partial v_c(x; b, x_0, v_0)}{\partial x} = 0, \tag{41}$$

which is:

$$-\frac{v_0\epsilon}{3}\left(3\,b^2+x^2\right)^{\frac{5}{6}}x_0^{\frac{2}{3}}\left(1+\epsilon\,(-x+x_0)\right)^{-\frac{2}{3}}\left(3\,b^2+x_0^2\right)^{-\frac{5}{6}}x^{-\frac{2}{3}}$$
$$+\frac{5\,v_0}{3}\sqrt[3]{1+\epsilon\,(-x+x_0)}x_0^{\frac{2}{3}}\sqrt[3]{x}\left(3\,b^2+x_0^2\right)^{-\frac{5}{6}}\frac{1}{\sqrt[6]{3\,b^2+x^2}} \tag{42}$$
$$-\frac{2\,v_0}{3}\sqrt[3]{1+\epsilon\,(-x+x_0)}\left(3\,b^2+x^2\right)^{\frac{5}{6}}x_0^{\frac{2}{3}}\left(3\,b^2+x_0^2\right)^{-\frac{5}{6}}x^{-\frac{5}{3}}=0$$

The solution for $x$ of the above minimum determines the jet's length, $x_j$,

$$x_j=\frac{4\,b^2\epsilon^2+\epsilon^2x_0^2+\sqrt[3]{D_2}\epsilon\,x_0+D_2^{\frac{2}{3}}+2\,\epsilon\,x_0+\sqrt[3]{D_2}+1}{4\,\epsilon\,\sqrt[3]{D_2}}, \tag{43}$$

where:

$$\begin{aligned}D_1 &= -16\,b^4\epsilon^4+429\,b^2\epsilon^4x_0^2-24\,\epsilon^4x_0^4+858\,b^2\epsilon^3x_0\\ &\quad-96\,\epsilon^3x_0^3+429\,b^2\epsilon^2-144\,\epsilon^2x_0^2-96\,\epsilon\,x_0-24,\end{aligned} \tag{44}$$

and:

$$D_2=-42\,b^2\epsilon^3x_0+\epsilon^3x_0^3-42\,b^2\epsilon^2+3\,\epsilon^2x_0^2+2\,b\sqrt{D1}\epsilon+3\,\epsilon\,x_0+1. \tag{45}$$

Figure 6 shows $x_j$ numerically.

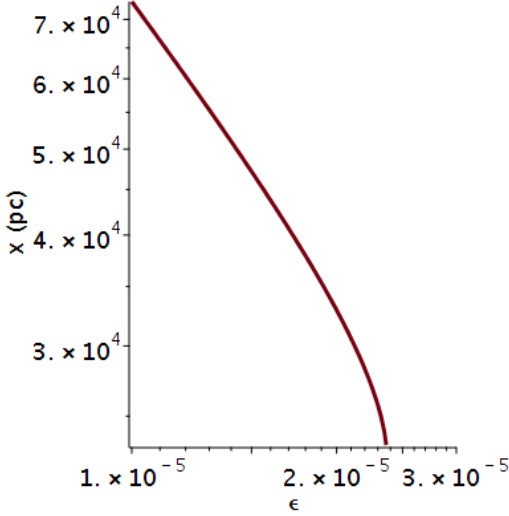

**Figure 6.** Length of the jet, $x_j$, in pc, as a function of $\epsilon$, with $b$ as in Table 1.

## 3. Conservation of the Relativistic Flux of Energy

The corrections in special relativity (SR) for stable atomic clocks in satellites of the Global Positioning System (GPS) are applied to satellites that are moving at a velocity of $\approx 3.87\ \frac{km}{s}$; see [31,32].

In astrophysics, we deal with velocities near that of light, and therefore, we should introduce relativistic conservation laws. The conservation of the relativistic flux of energy in SR in the presence of a velocity $v$ along one direction states that:

$$A(x)\frac{1}{1-\frac{v^2}{c^2}}(e_0+p_0)v=cost \tag{46}$$

where $A(x)$ is the considered area in the direction perpendicular to the motion, $c$ is the speed of light, $e_0=c^2\rho$ is the energy density in the rest frame of the moving fluid, and $p_0$ is the pressure in the rest

frame of the moving fluid; see Formula (A31) in [11,19]. In accordance with the current models of classical turbulent jets, we insert $p_0 = 0$. Then, the conservation law for the relativistic flux of energy is:

$$\rho c^2 v \frac{1}{1 - \frac{v^2}{c^2}} A(x) = cost. \tag{47}$$

In the presence of a Lane–Emden ($n = 5$) density profile, as given by Equation (22) and $A(x)$ as given by Equation (26), the conservation of relativistic flux of energy for a straight jet takes the form:

$$\frac{\rho_0 c^3 \beta \, \pi \, x^2 \, (\tan(\alpha/2))^2}{\left(1 + \frac{1}{3} \frac{x^2}{b^2}\right)^{5/2} (1 - \beta^2)} = \frac{\rho_0 c^3 \beta 0 \, \pi \, x_0{}^2 \, (\tan(\alpha/2))^2}{\left(1 + \frac{1}{3} \frac{x_0{}^2}{b^2}\right)^{5/2} \left(1 - \beta 0^2\right)}, \tag{48}$$

where $v$ is the velocity at $x$, $v_0$ is the velocity at $x_0$, $\beta = \frac{v}{c}$, and $\beta_0 = \frac{v_0}{c}$. The solution for $\beta$ to first order is:

$$\beta(x; x_0, b, \beta_0) = \frac{N}{\left(1 + \frac{1}{3} \frac{x_0{}^2}{b^2}\right)^{5/2} \left(\beta 0^2 - 1\right)}, \tag{49}$$

where:

$$
\begin{aligned}
N \;=\; & 9\sqrt{3\,b^2 + x_0{}^2}\,x^2 b^4 \beta_0{}^2 + 6\sqrt{3\,b^2 + x_0{}^2}\,x^2 b^2 x_0{}^2 \beta_0{}^2 + \sqrt{3\,b^2 + x_0{}^2}\,x^2 x_0{}^4 \beta_0{}^2 \\
& -9\sqrt{3\,b^2 + x_0{}^2}\,x^2 b^4 - 6\sqrt{3\,b^2 + x_0{}^2}\,x^2 b^2 x_0{}^2 - \sqrt{3\,b^2 + x_0{}^2}\,x^2 x_0{}^4 \\
& +\Bigg( 243\,x^4 (b^2 + \tfrac{1}{3} x_0{}^2)^5 \beta_0{}^4 + (-2\,x^4 x_0{}^{10} - 30\,b^2 x^4 x_0{}^8 - 180\,b^4 x^4 x_0{}^6 \\
& +(972\,b^{10} + 1620\,b^8 x^2 + 540\,b^6 x^4 + 360\,b^4 x^6 + 60\,b^2 x^8 + 4\,x^{10})x_0{}^4 \\
& -810\,b^8 x^4 x_0{}^2 - 486\,b^{10} x^4)\beta_0{}^2 + 243\,x^4 (b^2 + \tfrac{1}{3} x_0{}^2)^5 \Bigg)^{1/2}.
\end{aligned}
\tag{50}
$$

The equation for the relativistic trajectory is:

$$\int_{x_0}^{x} \frac{1}{\beta(x; x_0, b, \beta_0)\,c}\,dx = t. \tag{51}$$

The integral in the above equation does not have an analytical solution and should be integrated numerically. In order to have analytical results, two approximation are now introduced. The first approximation computes a truncated series expansion for the integrand of the integral in Equation (51), which transforms the relativistic equation of motion into:

$$F(x) - F(x_0) = t, \tag{52}$$

with:

$$F(x) = \frac{NF}{162\,x_0{}^2 \beta_0 b^{10} c}, \tag{53}$$

where:

$$
\begin{aligned}
NF \;=\; & (b^2)^{5/2}\,x\Big( 9\sqrt{3}\sqrt{3\,b^2 + x_0{}^2}\,b^4 \beta_0{}^2 x^2 + 6\sqrt{3}\sqrt{3\,b^2 + x_0{}^2}\,b^2 \beta_0{}^2 x^2 x_0{}^2 \\
& +\sqrt{3}\sqrt{3\,b^2 + x_0{}^2}\,\beta_0{}^2 x^2 x_0{}^4 - 9\sqrt{3}\sqrt{3\,b^2 + x_0{}^2}\,b^4 x^2 \\
& -6\sqrt{3}\sqrt{3\,b^2 + x_0{}^2}\,b^2 x^2 x_0{}^2 - \sqrt{3}\sqrt{3\,b^2 + x_0{}^2}\,x^2 x_0{}^4 - 162\sqrt{b^{10} x_0{}^4 \beta_0{}^2} \Big).
\end{aligned}
\tag{54}
$$

In the above analytical result, we have the time as a function of the distance (see Figure 7 with parameters as in Table 2 ), where the percentage error at $x = 15$ kpc is $\delta = 15.91\%$.

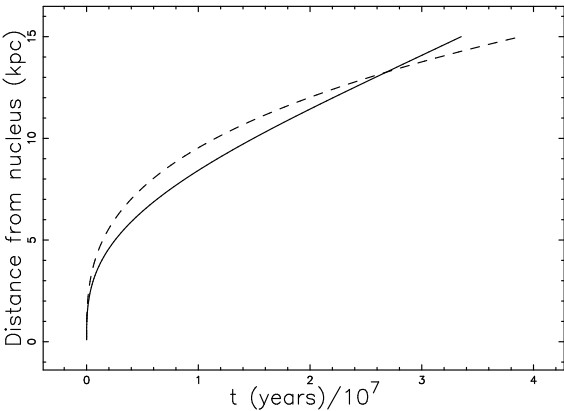

**Figure 7.** Numerical relativistic solution as given by Equation (51) (full line) and truncated series expansion as given by Equation (7) (dashed line), with parameters as in Table 2.

**Table 2.** Parameters for a relativistic extra-galactic jet.

| Parameter | Value |
|---|---|
| $x_0$ (pc) | 100 |
| $\beta_0$ | 0.9 |
| $b$ (pc) | 10,000 |

The second approximation computes a Padé approximant of order [2/1] (see [33–35]) for the integrand of the integral in Equation (51):

$$P(x) - P(x_0) = t, \tag{55}$$

with:

$$P(x)\frac{NP}{162\,b^{10}x_0{}^4\beta_0{}^2c}, \tag{56}$$

where:

$$
\begin{aligned}
NP(x) \;=\; & -x\,(b^2)^{5/2}\,x_0{}^2\beta_0\Big(9\,\sqrt{3}\sqrt{3\,b^2 + x_0{}^2}b^4\beta_0{}^2x^2 + 6\,\sqrt{3}\sqrt{3\,b^2 + x_0{}^2}b^2\beta_0{}^2x^2x_0{}^2 \\
& +\sqrt{3}\sqrt{3\,b^2 + x_0{}^2}\beta_0{}^2x^2x_0{}^4 - 9\,\sqrt{3}\sqrt{3\,b^2 + x_0{}^2}b^4x^2 - 6\,\sqrt{3}\sqrt{3\,b^2 + x_0{}^2}b^2x^2x_0{}^2 \\
& -\sqrt{3}\sqrt{3\,b^2 + x_0{}^2}x^2x_0{}^4 - 162\,\sqrt{b^{10}x_0{}^4\beta_0{}^2}\Big).
\end{aligned}
\tag{57}
$$

The above equation can be inverted, but the analytical expression for $x = G(t; x_0, \beta_0, b)$ as a function of time is complicated and is omitted here. As an example, with the parameters of Table 2, we have:

$$G(t) = \frac{NG}{DG}, \tag{58}$$

with

$$
\begin{aligned}
NG \;=\; & -2.9237 \times 10^{-17}\Big(-1.7397 \times 10^{54}\,t - 5.8851 \times 10^{56} \\
& +2.9816 \times 10^{20}\,\sqrt{1.9201 \times 10^{73} + 2.3032 \times 10^{70}\,t} \\
& +3.4042 \times 10^{67}\,t^2\Big)^{2/3} + 3.2399 \times 10^{21},
\end{aligned}
\tag{59}
$$

and:

$$
\begin{aligned}
DG \;=\; \Big( & -1.7397 \times 10^{54}\,t - 5.8851 \times 10^{56} \\[4pt]
& +2.9816 \times 10^{20}\,\sqrt{1.9201 \times 10^{73} + 2.3032 \times 10^{70}\,t + 3.4042 \times 10^{67}\,t^2}\,\Big)^{\frac{1}{3}}.
\end{aligned}
\tag{60}
$$

An example is shown in Figure 8, where the percentage error at $x = 15$ kpc is $\delta = 4.81\%$.

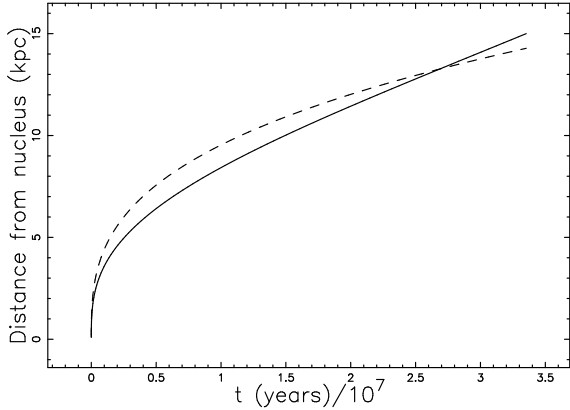

**Figure 8.** Numerical relativistic solution as given by Equation (51) (full line) and Padé approximant as given by Equation (8) (dashed line), with parameters as in Table 2.

*Relativistic Solution to Second Order*

We now suppose that the radiative losses are proportional to the relativistic flux of energy. The integral of the losses, $L_r$, between $x_0$ and $x$ is:

$$
L_r(x; x_0, \beta_0, b, c) = -\epsilon \frac{9\,(x - x_0)\,\rho 0\,c^3 \beta_0 x_0^2 \pi\,(\tan(\alpha/2))^2\,b^5 \sqrt{3}}{(3\,b^2 + x_0^2)^{5/2}\left(1 - \beta_0{}^2\right)}.
\tag{61}
$$

The conservation of the relativistic flux of energy in the presence of the back-reaction due to the radiative losses is:

$$
\frac{NR}{(3\,b^2 + x^2)^{5/2}\,(3\,b^2 + x_0^2)^{5/2}\,(\beta^2 - 1)\left(\beta_0{}^2 - 1\right)}
= \frac{9\,\rho 0\,\sqrt{3}c^3 \beta_0 x_0^2 b^5}{(3\,b^2 + x_0^2)^{5/2}\left(\beta_0{}^2 - 1\right)},
\tag{62}
$$

where:

$$
\begin{aligned}
NR \;=\; 81\,\rho 0\,b^5 \sqrt{3}\Big( & (b^2 + \tfrac{1}{3} x^2)^2 \epsilon\,(\beta + 1)\beta_0(\beta - 1)(x - x_0)x_0^2 \sqrt{3\,b^2 + x^2} \\[4pt]
& + (b^2 + \tfrac{1}{3} x_0^2)^2 \beta\,x^2(\beta_0 + 1)(\beta_0 - 1)\sqrt{3\,b^2 + x_0^2}\,\Big)c^3.
\end{aligned}
\tag{63}
$$

The solution of the above equation, to second order, for $\beta$ is:

$$
\beta = \frac{NB}{2\,(3\,b^2 + x^2)^{5/2}\,(\epsilon\,x - \epsilon\,x_0 - 1)\,(3\,b^2 + x_0^2)\,x_0^2 \beta_0},
\tag{64}
$$

where:

$$
\begin{aligned}
NB \;=\; & -\sqrt{3\,b^2 + x_0{}^2}\Bigg(\sqrt{3\,b^2 + x_0{}^2}\times \\
& \Big(x^4(\beta_0-1)^2(\beta_0+1)^2 x_0{}^{10} + 15\,b^2 x^4(\beta_0-1)^2(\beta_0+1)^2 x_0{}^8 \\
& + (972\,(b^2+\tfrac{1}{3}\,x^2)^5\beta_0{}^2\epsilon^2 + 90\,b^4 x^4(\beta_0-1)^2(\beta_0+1)^2)x_0{}^6 \\
& - 1944\,\epsilon\,(b^2+\tfrac{1}{3}\,x^2)^5\beta_0{}^2(\epsilon\,x - 1)x_0{}^5 + (972\,(b^2+\tfrac{1}{3}\,x^2)^5\beta_0{}^2 x^2\epsilon^2 \\
& - 1944\,(b^2+\tfrac{1}{3}\,x^2)^5\beta_0{}^2 x\epsilon + 4\,x^{10}\beta_0{}^2 + 60\,b^2 x^8\beta_0{}^2 + 360\,b^4 x^6\beta_0{}^2 \\
& + 270\,b^6(\beta_0{}^2+1)^2 x^4 + 1620\,b^8 x^2\beta_0{}^2 + 972\,b^{10}\beta_0{}^2)x_0{}^4 \\
& + 405\,b^8 x^4(\beta_0-1)^2(\beta_0+1)^2 x_0{}^2 + 243\,b^{10}x^4(\beta_0-1)^2(\beta_0+1)^2\Big)^{1/2} \\
& + 27\,(b^2+\tfrac{1}{3}\,x_0{}^2)^3(\beta_0+1)x^2(\beta_0-1)\Bigg).
\end{aligned}
\tag{65}
$$

The relativistic equation of motion with back-reaction can be solved by numerically integrating the relation in Equation (51). Figure 9 gives an example.

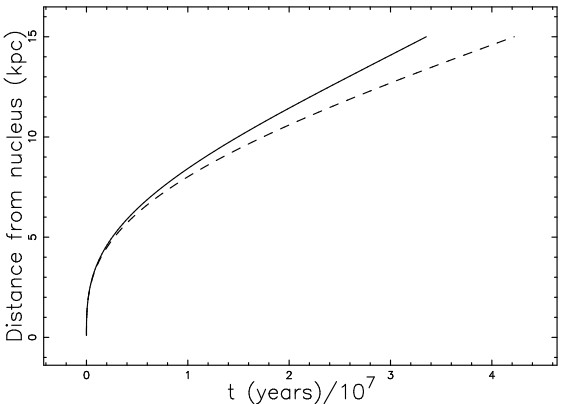

**Figure 9.** Numerical relativistic solution as given by Equation (51) (full line) and the solution with back-reaction, i.e., to second order, (dashed line), with parameters as in Table 2 and $\epsilon = 2.0 \times 10^{-5}$.

## 4. Astrophysical Applications

We now analyze two models for the synchrotron emission along the jet.

### 4.1. Direct Conversion

The flux of observed radiation along the center of the jet, $I_c$, in the classical case is assumed to scale as:

$$
I_c(x; x_0, v_0, b, \epsilon) \propto \frac{L(x; x_0, v_0, b, \epsilon)}{x^2},
\tag{66}
$$

where $L$, the sum of the radiative losses, is given by Equation (38).

The above relation connects the observed intensity of radiation with the rate of energy transfer per unit area. In the relativistic case:

$$
I_c(x; x_0, \beta_0, b, c) \propto \frac{L_r(x; x_0, \beta_0, b, c)}{x^2},
\tag{67}
$$

where $L_r$ is given by Equation (61).

A statistical test for the the goodness of fit is the observational percentage of reliability, $\epsilon_{\text{obs}}$,

$$\epsilon_{\text{obs}} = 100\left(1 - \frac{\sum_j |I_{obs} - I_{theo}|_j}{\sum_j I_{theo,j}}\right). \tag{68}$$

In order to make a comparison with the observed profile of intensity, we chose 3C31; see Figure 8 in [7]; Figure 10 shows the theoretical synchrotron intensity, as well as the observed one.

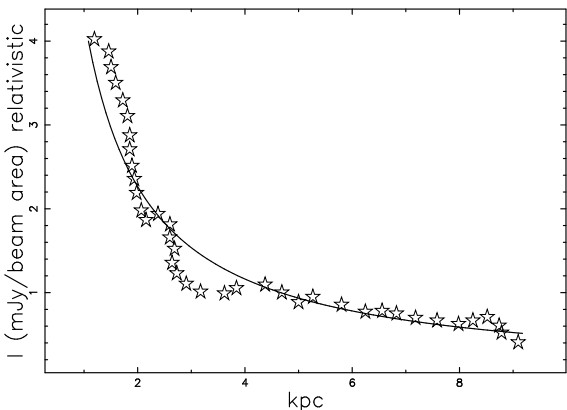

**Figure 10.** Observed intensity profile along the centerline of 3C31 (empty stars) and theoretical intensity as given by Equation (67), with parameters as in Table 2 (full line). The observational percentage of reliability is $\epsilon_{\text{obs}} = 86.19\%$.

### 4.2. The Magnetic Field of Equipartition

The magnetic field in centimeter-gram-second system of units (CGS) has an energy density of $\frac{B^2}{8\pi}$, where $B$ is the magnetic field. The presence of the magnetic field can be modeled assuming equipartition between the kinetic energy and the magnetic energy:

$$\frac{B(x)^2}{8\pi} = \frac{1}{2}\rho v^2. \tag{69}$$

Inserting the above equation into the classical equation for the conservation of the flux of energy (27), a factor of two will appear on both sides of the equation, leaving unchanged the result for the deduction of the velocity to first order. The magnetic field as a function of the distance $x$ when the velocity is given by Equation (29) and in the presence of a Lane–Emden ($n = 5$) profile for the density is:

$$B(x; x_0, b) = \frac{B_0 \left(3\, b^2 + x_0{}^2\right)^{\frac{5}{12}} x_0^{\frac{2}{3}}}{\left(3\, b^2 + x^2\right)^{\frac{5}{12}} x^{\frac{2}{3}}}. \tag{70}$$

where $B_0$ is the magnetic field at $x = x_0$. We assume an inverse power law spectrum for the ultrarelativistic electrons, of the type:

$$N(E)dE = KE^{-p}dE \tag{71}$$

where $K$ is a constant and $p$ the exponent of the inverse power law. The intensity of the synchrotron radiation has a standard expression, as given by Formula (1.175) in [36],

$$I(\nu) \approx 0.933 \times 10^{-23} \alpha_p(p) KlH_{\perp}^{(p+1)/2} \left(\frac{6.26 \times 10^{18}}{\nu}\right)^{(p-1)/2} \tag{72}$$

$$\text{erg}\,\text{s}^{-1}\text{cm}^{-2}\text{Hz}^{-1}\text{rad}^{-2}$$

where $\nu$ is the frequency, $H_\perp$ is the magnetic field perpendicular to the electron's velocity, $l$ is the dimension of the radiating region along the line of sight, and $\alpha_p(p)$ is a slowly-varying function of $p$, which is of the order of unity. We now analyze the intensity along the centerline of the jet, which means that the radiating length is:

$$l(x; \alpha) = x \tan\left(\frac{\alpha}{2}\right). \tag{73}$$

The intensity, assuming a constant $p$, scales as:

$$I(x; x_0, p) = \frac{I_0 B^{\frac{p}{2}+\frac{1}{2}} x}{B_0^{\frac{p}{2}+\frac{1}{2}} x_0}, \tag{74}$$

where $I_0$ is the intensity at $x = x_0$ and $B_0$ the magnetic field at $x = x_0$. We insert Equation (70) in order to have an analytical expression for the centerline intensity:

$$I(x; x_0, p, b) = \left(3 b^2 + x_0^2\right)^{\frac{5p}{24}+\frac{5}{24}} i_0 x^{-\frac{p}{3}+\frac{2}{3}} x_0^{\frac{p}{3}-\frac{2}{3}} \left(3 b^2 + x^2\right)^{-\frac{5p}{24}-\frac{5}{24}}. \tag{75}$$

The above equation for the intensity is relative to the unit area; in order to have the intensity on the centerline, $I_c$, we should make a further division by the area of interest, which scales $\propto x^2$:

$$I_c(x; x_0, p, b) = \frac{I(x; x_0, p, b)}{x^2}. \tag{76}$$

Figure 11 shows the theoretical synchrotron intensity with the variable magnetic field, as well as the observed one for 3C31.

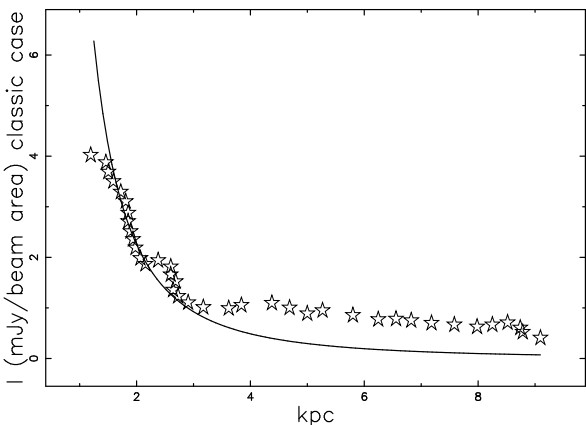

**Figure 11.** Observed intensity profile along the centerline, $I_c$, of 3C31 (empty stars) and theoretical intensity as given by Equation (76), with parameters as in Table 1. The observational percentage of reliability is $\epsilon_{\mathrm{obs}} = 73.79\%$.

## 5. Conclusions

Classical case: The approximate trajectory of a turbulent jet in the presence of a Lane–Emden ($n = 5$) medium has been evaluated to first order; see Equation (35). The solution for the velocity to first order allows the insertion of the back-reaction, i.e., the radiative losses, in the equation for the flux of energy conservation (see Equation (39)) and, as a consequence, the velocity corrected to second order (see Equation (40)). The trajectory, calculated numerically to second order, is shown in Figure 5. The radiative losses allow evaluating the length at which the advancing velocity of the jet is zero. This length has a complicated analytical expression and was presented numerically; see Figure 6.

Relativistic case: In the relativistic case, it is possible to derive an analytical expression for $\beta$ to first order (see Equation (49)) and to second order (taking into account radiative losses) (see Equation (64)).

The relativistic trajectory to first order has been evaluated through a series (see Equation (52)) or a Padé approximant of order [2/1] (see Equation (58)). The relativistic equation of motion to second order (back-reaction) has been evaluated numerically; see Figure 9. In other words, with the introduction of the radiative losses, the length of the classical or relativistic jet becomes finite, rather than infinite.

An astrophysical application: The radiative losses are represented by Equation (37) in the classical case and by (61) in the relativistic case. A division of the two above quantities by the area of interest allows deriving the theoretical rate of energy transfer per unit area, which can be compared with the intensity of radiation along the jet, for example 3C31; see Figure 10. The spatial behavior of the magnetic field is introduced under the hypothesis of equipartition between the kinetic and magnetic energy (see Equation (70)), and this allows closing the standard equation for the synchrotron emissivity (see Equation (73)).

**Funding:** This research received no external funding.

**Conflicts of Interest:** The author declares no conflict of interest.

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
