# Peer review of "Classical and Relativistic Evolution of an Extra-Galactic Jet with Back-Reaction"

_galaxies, doi:10.3390/galaxies6040134_

Round 1
Reviewer 1 Report
Referee report: « Classical and relativistic evolution of an extra-galactic jet with the back-reaction »
This work addresses the trajectory of turbulent astrophysical jet and influence of radiative cooling in it by mean of analytical method. I would recommend publication of this paper following a consideration of my comments below that I believe could improve this paper.
A- Introduction
1) The author indicates that the total energy available in the jet decrease due to the radiative loss. However, for relativistic radio-jet the energy flux is of order of 1036 -1039W and radiative energy loss remain very small 1028 W. Could the author explain more how this energy loss effect the jet dynamics. The author could explain more this contribution using the figure 2 and 3 of the paper.
B- Section 2
1) Could the author why the turbulent jet has the same density of the surrounding medium.
2) In the subsection 2.2, the author should give a reference for the energy conservation equation (2).
3) In the equation (2) the thermal pressure is neglected. The author should explain why.
4) The definition of the velocity in the equation (2) is it relativistic or classical.
5) Could author give the relation between the distance from the center “r” and the position “x”
6) In subsection 2.3, could the author draw the velocity evolution when epsilon=0
7) Could author explain more the figure 4. Indeed, the author should explain why the jet length drop with epsilon and what is the relation with external medium pressure.
C- Section 3
1) The use of the equation (2) for the velocity in the equation (17) is possible only if the definition (2) is relativistic. Could author demonstrate it.
2) The author should explain difference between the classical and relativistic case.
D- Section 4
In the astrophysical application, the author could deduce the radiative cooling effect in astrophysical jet 3C31 and thus gives the energy loss fraction.
Author Response
In the following there is the answer to the referee's report
organized by points
A- Introduction
Point 1 of the referee)
The author indicates that the total energy available in the jet decrease due to the radiative loss. However, for relativistic radio-jet the energy flux is of order of 1036 -1039W and radiative energy loss remain very small 1028 W. Could the author explain more how this energy loss effect the jet dynamics. The author could explain more this contribution using the figure 2 and 3 of the paper.
Answer to point 1
The paper deals with the flux of kinetic energy at the jet's place and therefore the observed luminosity at earth is not here considered. Now the introduction starts with the Fanaroff-Riley classification.
B- Section 2
Point of the referee 1)
Could the author why the turbulent jet has the same density of the surrounding medium.
Answer to point 1)
A new subsection 2.1 on turbulent jets has been inserted where the existing knowledge is reviewed.
Point of the referee 2)
In the subsection 2.2, the author should give a reference for the energy conservation equation (2).
Answer to point 2)
The conservation of energy flux is now better explained see new red text at the beginning of section 2.3
Point of the referee 3)
In the equation (2) the thermal pressure is neglected. The author should explain why.
Answer to point 3)
The pressure is absent in theory of the turbulent jets as outlined in point 2 after formula (4)
Point of the referee 4)
The definition of the velocity in the equation (2) is it relativistic or classical.
Answer to point 4)
The title of subsection now contains the world Classical.
Point of the referee 5)
Could author give the relation between the distance from the center 'r' and the position 'x'.
Answer to point 5)
Now new equations (25) and (26) relate r with x.
Point of the referee 6)
In subsection 2.3, could the author draw the velocity evolution when epsilon=0.
Answer to point 6)
Now Figure 2 contains the case epsilon=0
Point of the referee 7)
Could author explain more the figure 4. Indeed, the author should explain why the jet length drop with epsilon and what is the relation with external medium pressure.
Answer to point 7)
Now the discussion on the minimum for velocity has been enlarged , see red section after equation (41). Impossible to comment with evolution of the pressure which in the turbulent jet is zero.
C- Section 3
Point of the referee 1)
The use of the equation (2) for the velocity in the equation (17) is possible only if the definition (2) is relativistic. Could author demonstrate it.
Answer to point 1)
There was a LATEX error in the previous manuscript and the density profile was without number of equation. By the way the new subsection 2.2 introduces the Lane_Emden profile. Now the relativistic equation is introduced more clearly and the reference to the adopted profile is correct.
Point of the referee 2)
The author should explain difference between the classical and relativistic case.
Answer of the author to point 2)
I have inserted some comments on the velocity of transition from classical physics to relativistic physics , see first paragraph of Section 3
D- Section 4
Point of the referee)
In the astrophysical application, the author could deduce the radiative cooling effect in astrophysical jet 3C31 and thus gives the energy loss fraction.
Reviewer 2 Report
This manuscript is shown the new analytical expression for the velocity and approximate trajectory through the conservation of total energy in relativistic jets. The author considered back reaction due to radiative losses which gives new findings for trajectory. The manuscript contains enough new findings and contents. Therefore I recommend that this manuscript should be published in Galaxies.
But this manuscript is still incomplete. So the authors should address following issue.
Major comments:
1.The author should improve the introduction part by written more detail back ground of your research. What is the extra-galactic radio jets, which type of radio jets you are considered in this work, what is the turbulent jet, why such turbulent is developed? The authors should address these basic questions and provides more information.
2.In section 2, it is not enough information about your initial condition of jets. What kind of jet shape, velocity, and density profile do you consider?
3.In your work, you considered Lane-Emden density profile with n=5. Why such density profile do you consider? If it is supported by some observations or theory, you should provide the reference.
4.In you approach, you considered the conservation of the total flux of energy. But in the equation, it looks only considered kinetic energy. Why internal energy is neglected? And how about the magnetic field contribution?
5.In your work, hydrodynamics is considered. If you consider the magnetic field, how change the results? It would be nice to put some comments about magnetic field contribution.
6.In the relativistic jet case, radiate loss is not much important for the dynamics of jet propagation than Newtonian YSO jets although it is affected cooling of electrons. The author should give the comments for the why radiative loss is dynamically important for relativistic jet cases.
Author Response
The comments of the referee were interesting. In the following you will find the answer point by point. The new parts of the manuscript are in red.
Major comments:
Point of the referee 1)
The author should improve the introduction part by written more detail back ground of your research. What is the extra-galactic radio jets, which type of radio jets you are considered in this work, what is the turbulent jet, why such turbulent is developed? The authors should address these basic questions and provides more information.
Answer to point 1 )
The beginning of the introduction has been enlarged. A new subsection on turbulent jets has been introduced. see subsection 2.1.
Point of the referee 2)
In section 2, it is not enough information about your initial condition of jets. What kind of jet shape, velocity, and density profile do you consider?
Answer to point 2)
Now Section 2 contains two new subsections and the basic assumptions should be better understood.
The initial parameters are reported in Table I and for sake of simplicity I considered only one profile of density.
Point of the referee 3)
In your work, you considered Lane-Emden density profile with n=5. Why such density profile do you consider? If it is supported by some observations or theory, you should provide the reference.
Answer to point 3)
Now a special subsection 2.2 is dedicated to the Lane-Emden density profile with n=5 which is now better defined. The above profile is an assumption of the theory here presented.
Point of the referee 4)
In you approach, you considered the conservation of the total flux of energy. But in the equation, it looks only considered kinetic energy. Why internal energy is neglected? And how about the magnetic field contribution?
Answer to point 4)
The word total in now disappeared in connection with word flux. The canonical theory of turbulent jets is developed with pressure equal to zero and without magnetic field; I am following these prescriptions. Further on the new subsection 4.2 analyzes the evolution of the magnetic field of equipartition.
Point of the referee 5)
In your work, hydrodynamics is considered. If you consider the magnetic field, how change the results? It would be nice to put some comments about magnetic field contribution.
Answer to point 5)
The magnetic field can be inserted in the theory of turbulent jets under some simplifying assumption. Here we assumed equipartition between magnetic energy and kinetic energy , see new subsection 4.2.
Point of the referee 6)
In the relativistic jet case, radiate loss is not much important for the dynamics of jet propagation than Newtonian YSO jets although it is affected cooling of electrons. The author should give the comments for the why radiative loss is dynamically important for relativistic jet cases.
Answer to point 6)
In the point relativistic case of the conclusions, I have inserted the following comment."In other words with the introduction of the radiative losses the lenght of the classical or relativistic jet becomes finite rather than infinite."
Round 2
Reviewer 1 Report
Dear Author
Author improve the paper, the context and application of the model well explain in the new version. I accepte the new version of the paper for publication
Best regards
Author Response
All OK
Reviewer 2 Report
The author addressed my issues and well revised the manuscript. But I still have several minor issues, which the author should address before it is published in Galaxies.
Minnor comments:
1. In Introduction, citation of Figure number is wrong. Please correct it.
2. In Figures 1 and 2, it should have reference or credit of figures.
Author Response
Point of the referee 1)
In Introduction, citation of Figure number is wrong. Please correct it.
Answer to point 1)
I have corrected the LATEX typo
Point of the referee 2)
2. In Figures 1 and 2, it should have reference or credit of figures.
Answer to point 2)
I have inserted the credit of the above figures in the caption.